# Airway Epithelial Cultures of Children with Esophageal Atresia as a Model to Study Respiratory Tract Disorders

**DOI:** 10.3390/children10061020

**Published:** 2023-06-05

**Authors:** Henriette H. M. Dreyer, Eleonora Sofie van Tuyll van Serooskerken, Lisa W. Rodenburg, Arnold J. N. Bittermann, Hubertus G. M. Arets, Ellen M. B. P. Reuling, Johannes W. Verweij, Eric G. Haarman, David C. van der Zee, Stefaan H. A. J. Tytgat, Cornelis K. van der Ent, Jeffrey M. Beekman, Gimano D. Amatngalim, Maud Y. A. Lindeboom

**Affiliations:** 1Department of Pediatric Pulmonology, Wilhelmina Children’s Hospital, University Medical Center, 3508 AB Utrecht, The Netherlandsg.d.amatngalim@umcutrecht.nl (G.D.A.); 2Regenerative Medicine Center Utrecht, University Medical Center, Utrecht University, 3584 CX Utrecht, The Netherlands; 3Department of Pediatric Surgery, Wilhelmina Children’s Hospital, University Medical Center Utrecht, 3508 AB Utrecht, The Netherlands; 4Pediatric Upper Gastrointestinal and Airway Treatment Center, Wilhelmina Children’s Hospital, University Medical Center Utrecht, 3508 AB Utrecht, The Netherlands; 5Department of Pediatric Otorhinolaryngology, Pediatric Wilhelmina Children’s Hospital, University Medical Center Utrecht, 3508 AB Utrecht, The Netherlands; 6Department of Paediatric Pulmonology, Emma Children’s Hospital, Amsterdam UMC, 1105 AZ Amsterdam, The Netherlands

**Keywords:** esophageal atresia, tracheal anomaly, airway epithelium, organoids, primary ciliary dyskinesia

## Abstract

Esophageal atresia (EA) is a rare birth defect in which respiratory tract disorders are a major cause of morbidity. It remains unclear whether respiratory tract disorders are in part caused by alterations in airway epithelial cell functions such as the activity of motile cilia. This can be studied using airway epithelial cell culture models of patients with EA. Therefore, the aim of this study was to evaluate the feasibility to culture and functionally characterize motile cilia function in the differentiated air–liquid interface cultured airway epithelial cells and 3D organoids derived from nasal brushings and bronchoalveolar lavage (BAL) fluid from children with EA. We demonstrate the feasibility of culturing differentiated airway epithelia and organoids of nasal brushings and BAL fluid of children with EA, which display normal motile cilia function. EA patient-derived airway epithelial cultures can be further used to examine whether alterations in epithelial functions contribute to respiratory disorders in EA.

## 1. Introduction

Esophageal atresia (EA) is a rare birth defect in which the esophagus is fused to the trachea or interrupted. Respiratory tract disorders, including chronic cough, recurrent respiratory tract infections, pneumonia or even life-threatening brief resolved unexplained events (BRUEs), are a major cause of long-term problems in these patients [1,2]. These symptoms are most common during infancy and childhood [1,3]. The majority of children with EA have restrictive and/or obstructive pulmonary function and a smaller lung capacity compared to healthy children [4,5,6,7]. However, the cause of respiratory morbidity in EA is not yet clearly understood.

Previous studies suggested functional abnormality of the tracheal airway epithelium in EA patients [5,8,9]. It is expected that cilia are lacking at the original location of the tracheo-esophageal fistula which may affect mucus clearance and cause colonization of microbes that are continuously inhaled, explaining recurrent respiratory tract infections. Similar to chronic respiratory diseases such as asthma and chronic obstructive pulmonary disease (COPD), environmental risk factors such as microbial infections may cause imprinted defects in airway epithelial cell functions, which persist in cell culture [10,11,12]. Airway epithelial cells of individuals with EA may also display impaired epithelial functions due to patient-specific genetic risk factors that are associated with congenital birth defects in EA [13]. For instance, genetic defects in the transcription factor SOX2 may lead to impaired separation of the foregut into the esophagus and trachea [14], while SOX2 has also been described as a key regulator of airway basal stem and progenitor cells. A potential mechanism affected in the airway epithelium of patients with EA is dysmotility of the cilia, which can impair the clearance of mucus from the respiratory tract. However, no previous studies have been conducted to assess ciliary function in the epithelial cells of individuals with EA.

We and others previously described methods for culturing airway epithelial cells from nasal brushings and BAL fluid from individuals with respiratory diseases such as cystic fibrosis (CF), asthma, or COPD [15,16,17,18,19]. These technical reports show that non-invasive methods such as nasal brushings or leftover materials from diagnostic tests can be used to isolate airway epithelium cells and to model patient-derived tissues in vitro. The cultured cells can then be used to study airway diseases or infections in a controlled and defined environment [20,21,22,23,24].

Here, we report for the first time that also airway epithelial cells of EA patients can be isolated and cultured by these protocols. We show that EA patient-derived airway epithelial cell cultures can be used to study alterations in epithelial cell function and morphology. We, therefore, evaluated the feasibility to culture airway epithelial cells derived from nasal brushings and BAL fluid of patients with EA in mucociliary differentiated air–liquid interface (ALI) cultures and 3D airway organoids. As a proof-of-concept of studying airway epithelial functions, we furthermore examined motile cilia activity in differentiated airway epithelial cell cultures of patients with EA, which was compared to healthy individuals and a patient with primary ciliary dyskinesia (PCD).

## 2. Materials and Methods

### 2.1. Human Materials and Informed Consent

An observational study was conducted including seven EA patients (Table 1) that underwent general anesthesia and had an indication for bronchoalveolar lavage (BAL) between July and December 2019. Patients were considered eligible for this study if they had any type of esophageal atresia and they had to undergo general anesthesia. Furthermore, nasal brushings from a healthy individual (age 32, no symptoms) and a patient (age 23, mild respiratory symptoms) with PCD were included as reference samples. Informed consent from the parents was obtained and entailed. This study was approved by the Ethical Board for the use of Biobanked materials TcBIO (Toetsingscommissie Biobanks Utrecht, The Netherlands), an institutional Medical Research Ethics Committee of the University Medical Center Utrecht (protocol number: 19/763; approved on 29 May 2020). All experimental procedures were conducted between June 2020 and August 2022.

### 2.2. Isolation and Differentiation of Airway Cells

Nasal brushings were collected with a cytology brush (C0004, CooperSurgical, Trumbull, CT, USA) under general anesthesia as previously described by Rodenburg et al. [15] Different sizes of brushes were available, depending on the age of the patient. The brush was rinsed with phosphate-buffered saline (PBS), prior to insertion of the nostril and the brush was inserted until a resistance was felt with rotating and linear movements. The brush was removed from the nostril and placed in a tube containing 5 mL of cell culture medium with antibiotics (Advanced DMEM/F12 (12634-028, Thermo Fisher Scientific, Waltham, MA, USA) with GlutaMax (35050-061, Thermo Fisher Scientific), Hepes (15630080, Thermo Fisher Scientific), Penicillin-Streptomycin (15070-063, Thermo Fisher Scientific), and primocin (ant-pm-2, InvivoGen, San Diego, CA, USA)) [15]. A nasal brushing was performed from both the left and right nostrils. The bronchoalveolar lavage (BAL) was carried out by the anesthesiologist after the nasal brushings were performed. Normal sterile saline (0.9% NaCL solution), ranging from 5–20 mL, was flushed through the endotracheal tube and was retrieved by mechanical aspiration. Half of the collected BAL sample was used for diagnostic purposes. The leftover material was used for airway epithelial stem cell isolation within four hours after sampling. Nasal and bronchial airway epithelial cells were isolated, expanded, and differentiated as previously described [15]. In short, airway basal stem cells were isolated by making a single cell suspension of the pellet yielded from nasal brush or BAL sample. After expansion, the stem cells were differentiated in ALI cultures using Transwell^®^ inserts (3470, Corning, Corning, NY, USA) for at least 18 days. Airway organoids were generated from epithelial fragments of a differentiated ALI culture as previously described [15].

### 2.3. Fixation and Immunofluorescent Microscopy

Differentiated nasal and bronchial ALI cultures (*n* = 3) were fixed and stained as previously described [25]. Primary antibodies (β-tubulin IV (MU178-UC, Emergo Biogenex, Fremont, CA, USA), MUC5AC (ab198294, Abcam, Cambridge, UK), were incubated for 1 h at RT. Afterward, secondary antibodies (A-21240 and A11034, Invitrogen, Waltham, MA, USA), together with phalloidin (A34055, Mol. Probes, Eugene, OR, USA) and DAPI (D9542, Sigma, St. Louis, MO, USA), were added for 30 min. Both primary and secondary antibodies were diluted 1:500 in blocking buffer (1% BSA + 0.3 Triton X-100 (T8787, Sigma-Aldrich, St. Louis, MO, USA) in PBS). Images were acquired with a Zeiss LSM800 confocal microscope (40× objective). Image quantification was performed using Fiji (Max Planck Institute, Dresden, Germany) and CellProfiler (Broad Institute, Cambridge, MA, USA). Cultures did not show contamination with other cell types such as fibroblasts.

### 2.4. Analysis of Ciliary Beat Frequency (CBF)

Ciliary beat frequency (CBF) was determined in differentiated ALI cultures and organoids by high-speed video microscopy (HSVM) on a Thunder Imager 3D live Cell using a DFC9000 GTC camera (Leica). ALI cultures were imaged in phase contrast (40× dry objective) and 3D organoids were captured in brightfield (40× dry objective) at 37 °C and 5% CO_2_. Videos were recorded at 203 frames per second (fps) for 512 frames in total or 404 fps for 1024 frames. For ALI filters, five different locations with moving cilia were selected for each video and beating was determined twice for two seconds. For determination of the wave pattern in organoids, cilia on different cells per condition were observed to validate the presence of an effective and recovery stroke. The same cilia were followed twice for 1–2 s to determine CBF. CBF was performed randomized on coded videos.

## 3. Results

### 3.1. Characteristics of Patients with Esophageal Atresia

Patient characteristics are presented in Table 1. Three donors were female and four were male. The median age was 6.4 months. Five patients had esophageal atresia Type C (with a distal tracheo-esophageal fistula) and two patients had type A (esophageal atresia without fistula). Associated anomalies were present in five out of seven patients. All but one had respiratory symptoms; patient 2 (male, 4 days) is the only patient without any respiratory symptoms.

### 3.2. In Vitro Differentiation of EA Patient Derived Airway Cultures

We first determine the feasibility to isolate, expand, and differentiate nasal and bronchial airway epithelial cells from individuals with EA derived from nasal brushings and BAL fluid, respectively (Figure 1). Culturing of all nasal brushings and four out of seven BAL samples was successful. Analysis of differentiated ALI-HNEC (human nasal epithelial cells) and –BAL of EA patients by immunofluorescence imaging showed the presence of β-tubulin IV+ ciliated and MUC5AC+ goblet cells in both nasal as well as bronchial cultures after 18 days of differentiation (Figure 2A). Image quantification of differentiated cells observed at the cell surface showed on average 31.1% goblet cells and 39.8% ciliated cells in nasal cultures and 29.6% goblet cells and 42.8% ciliated cells in bronchial cultures (Figure 2B).

### 3.3. Ciliary Activity

To assess the cilia activity of the differentiated ALI cultures of EA patients, HSVM was performed after 18 days of differentiation (Figure 2C). Differentiated ALI cultures of nasal cells from a healthy individual and a PCD patient were used as positive and negative control, respectively. CBF of the healthy control was between 14–22 Hz (Figure 2D). Beat frequency of all EA donors was between 9–23 Hz with an average of 17.6 Hz in nasal cultures and 16.1 Hz in bronchial cultures (Figure 2E). No ciliary beating could be detected in the PCD donor.

To enable observation of ciliary movement from the lateral side, differentiated ALI cultures were converted into 3D airway organoids. Similar to ALI cultures, airway organoids displayed goblet and ciliated cells, as confirmed by immunofluorescence imaging (Figure 3A). The cilia wave pattern was assessed by eye and the presence of an effective and recovery stroke was confirmed in selected ciliated cells (Figure 3B). CBF in airway organoids of individuals with EA (6–19 Hz) was in a similar range as the CBF counted in the healthy donor (11.5–20 Hz) (Figure 3C). The nasal organoids of donors with EA had an average frequency of 11 Hz and bronchial organoids 12.7 Hz (Figure 3D). In contrast, nasal organoids from an individual with PCD did not show any ciliary movement.

## 4. Discussion

In this study, we showed that it is feasible to isolate and differentiate airway epithelial cells from nasal as well as bronchial airway samples from individuals with EA. We found a lower success rate of airway epithelial cells isolated from BAL samples compared to nasal brushings. We were able to isolate epithelial cells from 4 out of 7 BAL samples, due to recurrent infections and insufficient growth of isolated epithelial cells in three cases. Upon successful isolation, we were able to differentiate both nasal and bronchial airway epithelial cells in ALI cultures, which displayed both ciliated and goblet cells. Moreover, we were able to culture 3D airway organoids derived from differentiated ALI –culture-derived epithelial fragments.

Nasal and bronchial culture models could be used to evaluate motile cilia activity. In both nasal and BAL-derived airway epithelial cell cultures from individuals with EA, we observed a similar beat frequency and wave pattern compared to a healthy control subject, which was in contrast to the ciliary wave pattern of an individual with PCD. This suggests that there was no primary ciliary defect present in the EA patients, as proposed by Engeseath et al. [26].

The use of leftover materials from diagnostic tests and the isolation of epithelial cells from nasal brushings in combination with efficient protocols for in vitro cell differentiation are a non-invasive alternative for when lung tissue cannot be obtained and should be considered for future studies. For example, the mechanism of secondary ciliary dyskinesia can be further studied by in vitro stimulation experiments such as effects of respiratory infection in cell cultures from individuals with EA. Recent studies showed that respiratory cell cultures derived from various isolation techniques can be used to study airway diseases or infections in a controlled and defined environment (Table 2). In addition, it may be assessed in future studies whether other airway epithelial cell functions are affected in EA, such as epithelial barrier integrity, wound healing, mucus secretion, and the expression and release of pro-inflammatory mediators.

Our study was limited to a small number of donor materials. Future studies in a larger subgroup of individuals with EA may be required to gain further insight in potential ciliary defects. The lower success rate of culturing cells from BAL samples is probably caused by the nature of this sampling technique [16,22,27]. Blind washings in general result in a lower cellular load when compared to brushings of the epithelial surface, as conducted for nasal sampling. We have, however, included these samples because they were leftover materials for diagnostic tests, and no additional procedure needed to be conducted. How sampling of airway cells with different methods affects the impact of cell isolation should be investigated in more depth in future studies. None of the chosen sampling techniques target the epithelial tissue located at the prior tracheoesophageal fistula specifically. It is unknown whether the transition tissue consists of tracheal epithelium, esophageal epithelium or both. It might be that ciliary cells are lacking at the location of the prior TEF. Moreover, often a small dent is seen at the place of the TEF. This may result in a loss of the mucociliary transport function of the trachea. Future studies should examine the tissue located at the prior tracheoesophageal fistula in more detail by studying bronchial brushings of this specific area or by analyzing the tissue removed from the trachea during surgery.
children-10-01020-t002_Table 2Table 2Respiratory cell culture models.Isolation TechniqueRisk for DonorRespiratory ModelLiteratureLeftover tissue from operationNot invasiveCigarette smoke, lung cancer, Rhinovirus, RSV, SARS- CoV-2[16,17,20,23,28]Tracheal aspiratesNot invasiveCigarette smoke, homeostasis[18,23]Nasal brush/washMinimal invasiveAsthma, CF, COPD, PCD, RSV, SARS-CoV-2[15,21,24,27,29,30,31] Bronchial biopsy/brushingMedium invasiveAsthma, COPD, PCD, Rhinovirus, SARS-CoV-2[27,32,33,34,35]BALMedium invasiveAsthma, lung cancer, RSV[16,22,27]CF = cystic fibrosis; COPD = and chronic obstructive pulmonary disease; PCD = primary ciliary dyskinesia; RSV = respiratory syncytial virus.

## Figures and Tables

**Figure 1 children-10-01020-f001:**
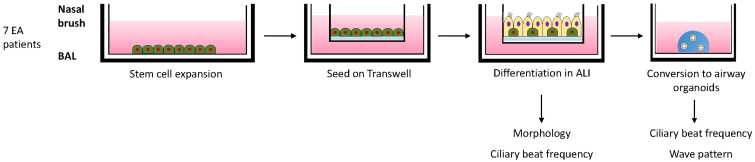
Schematic overview of culturing and analysis of differentiated airway epithelial cells of EA patients. Nasal brushings and BAL fluid from 7 EA patients were collected and airway epithelial cells were expanded in 2D cell cultures. After reaching confluence, the cells were seeded on Transwell inserts. After reaching confluency the apical medium was removed and cells were differentiated in an air-exposed condition. After 18 days of differentiation, cells were either imaged to determine the ciliary beat frequency (CBF) and fixed or converted into airway organoids. After a few days organoids were imaged to determine CBF and wave pattern.

**Figure 2 children-10-01020-f002:**
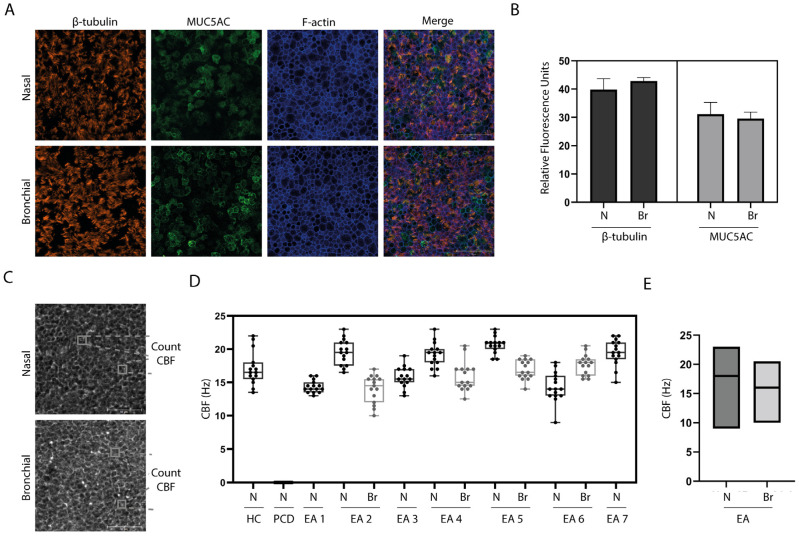
Characterization of ALI cultures from individuals with EA. (**A**) Immunofluorescence (IF) staining of differentiated ALI cultures of EA patients, generated from nasal (N) brushing and bronchial (Br) sample of randomly selected representative donor (Donor 2). β-tubulin IV is shown in orange, MUC5AC is shown in green and f-actin is shown in blue. (**B**) Quantification of IF staining of EA patients (*n* = 3) showing the relative fluorescence units. (**C**) Phase-contrast image of a nasal and BAL-derived ALI culture of randomly selected representative donor (Donor 2). (**D**) Ciliary beat frequency (CBF) determined in ALI cultures of a healthy control (HC) donor, an individual with primary ciliary dyskinesia (PCD) patient and 7 donors with esophageal atresia (EA). N = nasal; Br = bronchial. (**E**) Comparison of CBF in nasal versus bronchial ALI cultures from all EA patients.

**Figure 3 children-10-01020-f003:**
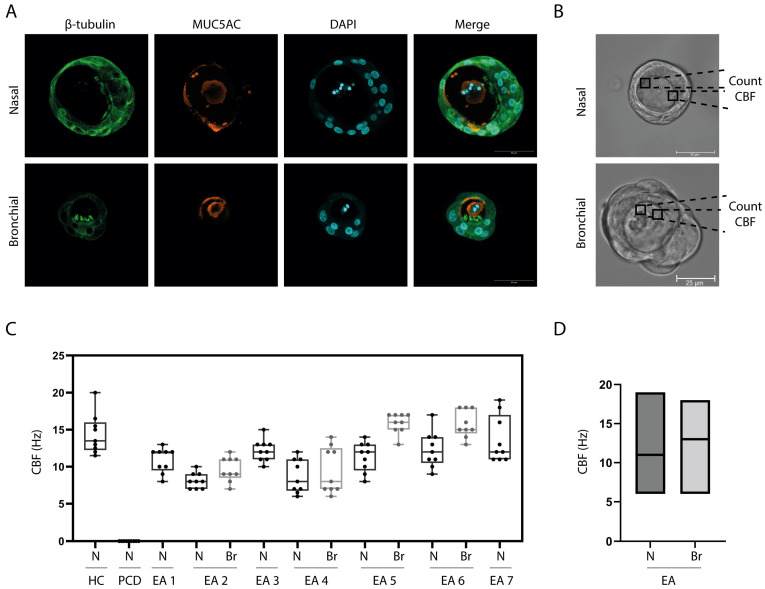
Ciliary beating in 3D airway organoids of patients with EA. (**A**) IF staining of organoids generated from a nasal brushing (Donor 1) and BAL sample (Donor 4). Images were randomly selected from representative donors. (**B**) Bright field image (40×) of a nasal organoid (Donor 1) and a BAL derived organoid (donor 4). Images were randomly selected from representative donors. (**C**) Ciliary beat frequency (CBF) determined in organoids from a healthy control (HC) donor, primary ciliary dyskinesia (PCD) patient, and all 7 donors with esophageal atresia (EA). N = nasal; Br= bronchial. (**D**) Comparison of CBF in nasal versus bronchial organoids from all EA patients.

**Table 1 children-10-01020-t001:** Characteristics of the included EA patients.

Gender, Age	Type EA	Associated Anomalies	Tracheomalacia	Respiratory History
1, female, 6.4 months	C	VACTERL	Yes, severe	No infections, mild respiratory symptoms
2, male, 4 days	A	Macrocefalie	None	No symptoms
3, male, 1.6 years	C	None	Yes, mild	Mild respiratory symptoms, cough
4, male, 4.8 years	C	None	Yes, mild	Multiple RTIs, cough
5, male, 1.2 months	C	VACTERL	Yes, mild	Mild respiratory symptoms, cough
6, female, 1 year	C	VACTERL	Yes, moderate	PPT, Cough
7, female, 7 days	A	Hypoplastic rib	None	Mild

VACTERL = vertebra, anal, cardiac, tracheo-esophageal, renal, limb; PPT = primary posterior tracheopexy; RTI = respiratory tract infections.

## Data Availability

The datasets used and/or analyzed during the current study are available from the corresponding author on reasonable request.

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
