# Peer review of "Airway Epithelial Cultures of Children with Esophageal Atresia as a Model to Study Respiratory Tract Disorders"

_children, 2023, doi:10.3390/children10061020_

Round 1

Reviewer 1 Report

Dreyer and colleagues present a nice manuscript, provided as a "Communication" to Children. The authors highlight the importance of respiratory epithelial cells/cultures obtained from patients suffering from e. atresia. They cultured the cells and established some technical methods, which should be taken into consideration when performing such kind of experiments. Therefor, the content is important and should be shared with the readership.

The manuscript is well written and easily to understand. However, there are some issues which have to be addressed during revision.

Why does the authors chose to submit a "communication"? The content of the manuscript can be strongly submitted either as an research article or technical manuscript. Please think about it!

Please add more technical manuscripts regarding nasal epithelial cell cultures to this manuscript, as these cells are important when considering stimulation experiments in vitro and can be alternatively taken without any significant risks when lung epithelial cells cannot be obtained. There are some important manuscripts published in 2020-2022 which should be adressed, as these information can also well implemented to this manuscript.

Please compare different cell culture models regarding your model in a table.

Please explain how the purity of the cells were analyzed.

Please write a comprehensive method section as there are no sufficient information regarding the kits, materials etc which are used.

Please provide a detailed discussion section and highlight the importance of such cell culture models for experiments.

Reviewer 2 Report

Thank you for the opportunity to review this manuscript, in which airway epithelial cells were obtained from nasal brushings and bronchoalveolar fluid of infants with esophageal atresia in order to evaluate the role of airway epithelial cell function in respiratory pathology in this condition. The manuscript demonstrates the feasibility of culturing airway epithelial cells and creating organoids in this condition, and found normal cilia motility, using a sample from a healthy airway and an airway of a person with primary ciliary dyskinesia as a positive and negative control respectively. This is a novel study, focusing on an important yet poorly understood area, of respiratory pathology in a relatively common (of the rare) congenital childhood conditions. As the authors correctly conclude, further investigation is required in this field. It is exciting to read this proof of concept study, however the discussion and conclusions should be expanded to further highlight the limitations here and to direct further research.

Specifically, the following should be highlighted and/or discussed further:

1. Further details regarding the positive and negative control samples is required. What were the ages of the patients from who they were obtained? What was their respiratory status (infection, antibiotics) at the time of culture?

2. Related to the above, please provide more information about the study participants. They are, for the most part, infants, and all have respiratory symptoms. It is not clear from Table 1 whether the symptoms were predominantly infective and/or related to retained mucous/impaired airway clearance, or mechanical - ie stridor, barking cough, which is frequently a baseline for children with this condition and unlikely related to epithelial cell function. The "other" and "musculoskeletal" anomalies listed in Table 1 for participants 2 and 7 should be further described if possible.  It is also important to report whether infective symptoms were present at the time of sampling, as it is established in other respiratory conditions that this in itself may impair epithelial cell function.

3. Images in the figures are selected from specific donors. It seems important to comment why these figures/donors were selected. Are these representative?

4. Bronchoalveolar samples were collected as blind washings via ETT. Typically, BAL samples are collected via bronchoscopy, which allows targeting of the area/bronchus of interest and often higher return. Does this explain the lower success rate of airway epithelial cell isolation from BAL fluid compared with nasal washings? The choice of technique needs to be discussed.

5. Bronchial brushings are also possible, and would allow targeting of the area of the TEF, which the authors postulate in the introduction to be a site of epithelial cell dysfunction. This should be further discussed.

Overall, this is an interesting proof of concept study expanding the application of airway epithelial cell culture and organoid development into a new area of pediatric respiratory medicine. As the authors note, the study and control population is too small to draw conclusions regarding pathophysiology. The discussion and conclusion should expand on the limitations and future directions, including the clinical applications and intepretation, as per suggestions above.

Round 2

Reviewer 1 Report

I‘m satisfied with the results. Good job! 

English grammar is okay.

Author Response

Reply to Academic Editor

Point 1: I‘m satisfied with the results. Good job! 

Response 1: Thank you very much.